# PhysioLLM: Supporting Personalized Health Insights with Wearables and Large Language Models

Cathy Mengying Fang[†], Valdemar Danry[†], Nathan Whitmore[†], Andria Bao[‡],
Andrew Hutchison[‡], Cayden Pierce[†], Pattie Maes[†],
[†]MIT Media Lab
{catfang, vdanry, nathanww, cayden, pattie}@media.mit.edu
[‡]MIT
{andria, aphutch}@mit.edu

*Abstract*—We present PhysioLLM, an interactive system that leverages large language models (LLMs) to provide personalized health understanding and exploration by integrating physiological data from wearables with contextual information. Unlike commercial health apps for wearables, our system offers a comprehensive statistical analysis component that discovers correlations and trends in user data, allowing users to ask questions in natural language and receive generated personalized insights, and guides them to develop actionable goals. As a case study, we focus on improving sleep quality, given its measurability through physiological data and its importance to general well-being. Through a user study with 24 Fitbit watch users, we demonstrate that PhysioLLM outperforms both the Fitbit App alone and a generic LLM chatbot in facilitating a deeper, personalized understanding of health data and supporting actionable steps toward personal health goals.

*Index Terms*—Large language model, Sleep, Conversational interface, Physiological data, Digital health app, Wearable, AI

## I. Introduction

The advent of wearable health monitors, such as Fitbit, Apple Watch, and Samsung Gear has made it possible to continuously collect detailed physiological data, such as heart rate, activity data, and sleep stages. They bring convenience and awareness to our personal health and provide a granular look into one's habits and how they affect physiology. These data and trends can help nudge healthier behavior and may even help detect health problems [1]. While it is important to make accessible and accurate health monitoring systems, individuals who wish to change their habits are currently required to first deeply understand their physiological data and how it correlates with their daily routine, and finally think of ways to work towards positive changes. However, users often struggle to make sense of the data and translate them into meaningful actions [2]. Interactions with the data are typically predefined by graphical user interfaces provided by the phone and wearables, which offer limited interaction and generic recommendations with few personalized insights.

Large Language Models (LLMs) potentially present a promising solution to these challenges. For one, they enable individuals to engage in unconstrained questioning and answering in natural language [3]. Second, they have the potential to relate health data and behaviors to a wealth of health literature [4]–[6]. Lastly, LLMs have a semantic understanding of the context that could grant flexibility in producing insights based on raw data [3]. Integrating LLMs with physiological data offers the potential to build systems that allow users to ask questions and receive personalized responses, enhancing their understanding of their health and motivating positive behavior changes. This research addresses two main questions: (1) how to implement an LLM-based system that generates personalized insights from physiological data and communicates them through natural language, and (2) how such a system impacts users' understanding of their data and helps them develop actionable health goals.

We designed PhysioLLM, a novel system that utilizes an orchestration of LLMs to deliver personalized insights by incorporating users' own data from already available wearable health trackers together with contextual information. Different from conventional health applications, our system conducts statistical analyses of the user's data to uncover patterns and relationships within the data. As a case study, we focus on improving sleep as the main health goal. Sleeping well is one the most important things to stay healthy physically and mentally [7]. The latest wearable devices offer in-depth reports on sleep, providing information on sleep timing, sleep stages and commonly used metrics such as wake time after sleep onset. They also typically provide a sleep score to indicate overall sleep quality. However, it is often not obvious to users how one can improve one's sleep score and the relationships between one's daytime activity and sleep.

To understand what might improve individuals' understanding of their data and what questions they might ask a conversational interface, we recruited actual users for an in-situ experiment. 24 adult Fitbit users shared their most recent week of Fitbit data. Each participant used a text-based chatbot that was either the complete PhysioLLM system with personal data and insights, an LLM chatbot with personal data but no access to insights, or a placebo off-the-shelf LLM chatbot with no personal data or generated insights. They filled out a survey before and after interacting with the interface that assessed

their understanding of their sleep data, how motivated they felt after interacting with the interface, and how actionable their goals were based on their interactions with the interface.

The results show that chatting with an LLM-based system, which provides effective personalized insights using our LLM architecture, improves one's understanding of their own health. The interface was perceived as more personalized than chatting with a generic LLM-based chatbot. In fact, the latter resulted in the user having less motivation to change, and their goals were found to be less actionable.

We also interviewed two sleep experts to review the personal insights generated by the system and its responses and suggestions provided to the user. The experts found the insights reasonable but noted the system's tendency to overemphasize correlation values. They suggested improving the system by providing the LLM with more background on the data generation process and tuning responses to be modest when based on sparse data and potentially spurious correlations.

In summary, the contributions of this work are:

- A novel orchestration of LLMs that integrates physiological and contextual data to support conversations about personalized health insights.
- An in-the-wild study with 24 users that interacted with the system and the study insights derived from quantitative and qualitative results.
- Evidences that show the interface is perceived as personalized and effectively improves users' understanding of their health through personalized insights.
- A preliminary valuation by two sleep experts of the accuracy and quality of the generated personal insights and suggestions.

## II. RELATED WORK

### A. LLMs For Health Prediction

The use of LLMs for medical tasks has rapidly increased, with applications such as knowledge extraction [8] and disease prediction [9]. Researchers found that the GPT-4 model exceeded the passing score on the United States Medical Licensing Examinations, an exam that allows individuals to practice medicine in the U.S., by over 20 points [10]. Med-PaLM2, a fine-tuned domain-specific medical LLM set a new state-of-the-art by scoring up to 86.5% on the MedQA dataset, a dataset containing expert answers to medical questions [4]. Meanwhile, researchers have finetuned LLMs for mental health specific tasks such as the prediction of stress and depression, achieving accuracies from 48% to 87% [11]. Taken together, these advancements highlight the substantial potential of LLMs in interpreting and reasoning about health information and their growing potential for supporting healthcare professionals. However, current approaches do not enable individuals who are not medical professionals to contextualize the knowledge with personal data and health goals. In contrast, PhysioLLM not only derives tailored insights from personal wearable health data but also allows the user to intuitively understand the implications of their data through conversations.

### B. LLM-based Data Analysis

Different from fine-tuning an LLM for domain-specific tasks, another approach is to prompt a large language model to generate code to then be run by a code executor to produce calculations and graphs [12]. While this approach has already found its way into commercial products[1], it requires explicit knowledge of the types of analyses to run. To overcome this challenge, other systems have added multiple "chains" or nodes of LLMs where each LLM in sequence selects the appropriate analysis from a set of possible analysis actions [13]. While this method enables users to explore their data without prior knowledge or conducting analyses themselves, it does not incorporate personal information about the user. Additionally, it still requires users to have some understanding of potential hypotheses to test based on data trends and to suggest these for further exploration. Physiollm takes into account the context of the personal health data and formulates hypotheses based on the data a priori. As such, it guides the user through a more focused conversation that prioritizes notable discoveries.

### C. LLM for Personal Health Insight Generation

Commercial systems are beginning to offer ChatGPT-based conversations to discuss training plans[2] and interpretations of heart rate variability data[3]. With the increasing availability of LLM-based services, prior research has emphasized the prediction accuracy of these models.

Many studies integrate personal health records from an electronic health record (EHR) for effective disease prediction [14] or to help patients understand health records [6]. Health-LLM proposed by Kim et al. adapts the public health prediction tasks with wearable data to enable personal health support [15]. Most related to our work is PH-LLM [5], a fine-tuned model for contextualizing physiological data and producing personalized insights. The work focuses on benchmarking the LLM's capability against human domain experts.

Stromel et al. compare the modality of the insight between text and chart and found LLM-generated text-based narrative to be more effective at helping people reflect on their data [16]. However, their investigation is limited to a one-turn interaction, and the data is limited to step count, whereas our system supports multi-turn conversations and explores the relationship among a variety of sensor data types to uncover relationships that may otherwise be difficult to see at a glance.

Frameworks like PH-LLM and Health-LLM focus on fine-tuning LLMs to achieve predictions, whereas PhysioLLM complements health wearables and apps by making the data more understandable and actionable. We study how an LLM-enabled system motivates specific healthy behaviors in real users, which is an area that lacks scientific investigation.

## III. MOTIVATION AND DESIGN GOALS

We hypothesize that engaging in a **personalized** conversation that includes **actionable insights** about one's health

---

[1]https://github.com/features/copilot
[2]https://www.whoop.com/
[3]https://welltory.com/

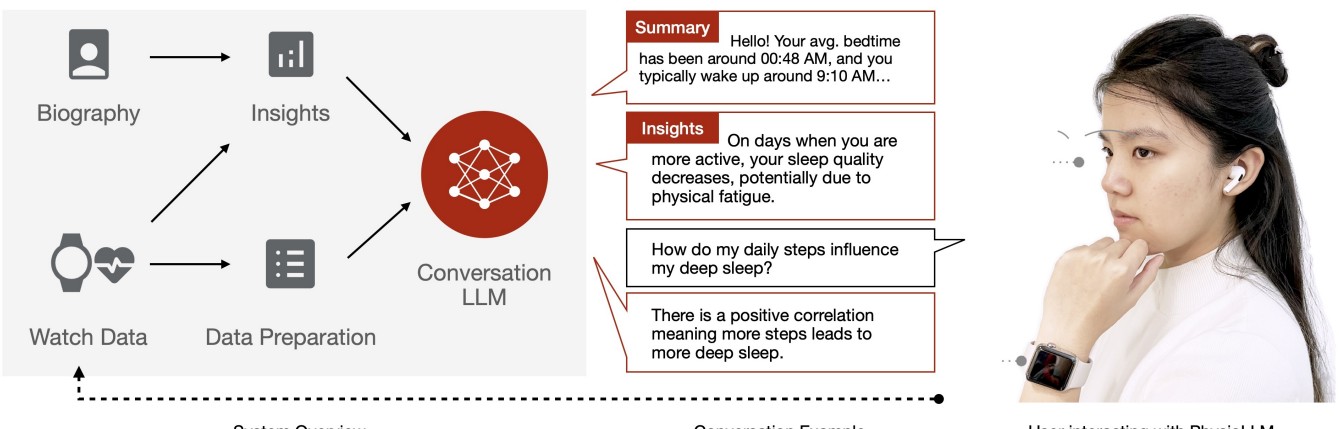

**System Overview**       **Conversation Example**       **User interacting with PhysioLLM**

Fig. 1. Overview of the PhysioLLM system with an example conversation.

data can enhance understanding of the data and the ability to develop effective action plans towards healthy behaviors. The concept of *personalization* is evident through the LLM's grounded knowledge of the user's data and its references to the data sources in its response. *Actionable insights* refer to the LLM-generated discoveries of trends, correlations, and patterns within the user's data, as well as actionable, follow-up questions and suggestions based on these discoveries. While current accompanying apps of wearable devices allow users to explore the collected data through graphical representations, uncovering actionable insights remains challenging. Data visualizations alone can lead to bias in interpreting their data, and one way to reduce such bias is to incorporate statistical analysis for comparison and correlation [17]. Additionally, although users can search for solutions to specific problems, these queries are often not contextualized within their data.

In addition to making personalized and insightful responses our primary research and design goal, we designed our system with the following important principles in mind: **Privacy-preserving**: To safeguard user confidentiality and trust, we ensure that no identifiable information is included in the communication with third-party systems; **Responsible**: To maintain ethical standards and avoid potential harm, our system should never provide medical or clinical diagnoses; **Accuracy**: To provide reliable and trustworthy information, we ensure all responses are based on the data sources and avoid any fabrication or hallucination of values; **Responsive**: To create a smooth and engaging user experience, the system is designed for fast response times, making the conversation feel seamless and fluid.

## IV. PHYSIOLLM ARCHITECTURE AND IMPLEMENTATION

The system (Figure 1) consists of three main components: data preparation, insight generation, and the conversational interface. Next, we describe each component in depth.

### A. Data Preparation

The quality of the responses depends on the quality and interpretability of the input data, which necessitates a process that prepares the data in formats that LLMs expect and instructs the LLMs on how to interpret the data. Initially, we thought to leverage the code-generation capabilities of LLMs

to provide real-time analysis of the data. Early experiments showed that this approach fails to be consistently accurate and fast, which are two important design principles. In addition, the need to generate bespoke functions is rare; meaningful analyses are often in the category of fundamental statistical analysis, such as mean, variance, trends over time, and correlation between data types. Thus, the system consists of an "offline" (as opposed to real-time) preparation phase that conducts statistical analysis on and summarizes the user's data. Specifically, the process is as follows:

**Data Filtering and Alignment**. Fitbit data was exported via "Google Data Takeout" and processed locally. The data was first filtered for the dates of interest. Raw data from different sensors have varying sampling rates. For example, *step count* is sampled every minute, *heart rate* is sampled every 5 minutes, and *sedentary minute* is sampled daily. We consolidated daily values for each data type and hourly values for *step count* and *heart rate*. Accurate representation of temporal information is essential, as the subsequent steps that derive the correlations and potential causal relationships rely on the temporal dimension. We aligned the different sensor data based on date and time considering the device's timezone. As we are interested in the effect of daily activities on sleep quality, we adjusted the "date of sleep" to correspond with the day following the recorded daytime activities. For simplicity, we excluded naps (i.e., not the main sleep event). In the event of missing data, an average of the weekly value was used.

**Generation of Summary, Trends, and Correlations**. After the data had been filtered and aligned, we summarized the data to extract the averages of the week, dates of min and max values, and trends. For trends, we used a permissive threshold of $\pm0.15$ because the goal is not to perform statistical hypothesis testing but rather to provide the LLM with narrative descriptions of possible trends. The hourly step count and heart rate were plotted to show the visual pattern of one's activity and heart rate each day over a week. Then, we calculated pairwise correlation values. An example of the pattern graph and correlation matrix plot is shown in Figure 2.

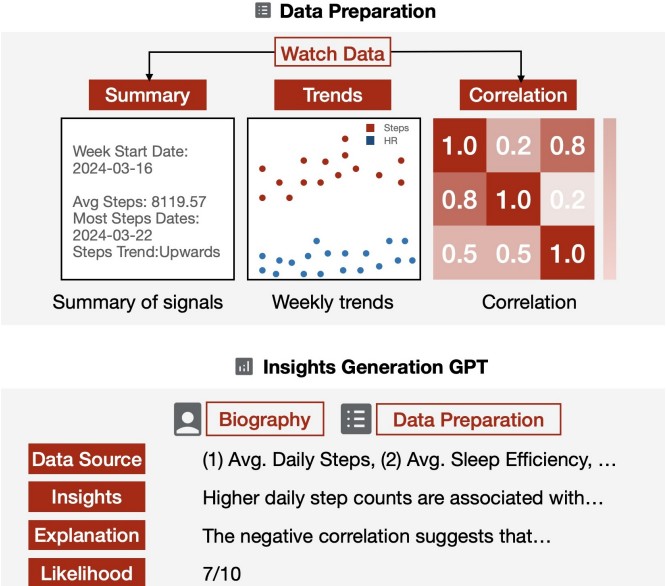

Fig. 2. The steps taken to summarize the data and generate insights from the data before the information is passed to the conversation LLM.

## B. User Modeler and Insight Generation

Deeper insights such as how the data correlate with each other and the implications of the data are not apparent to a user. As such, the mere integration of the user's data in an LLM is not enough. In addition, while general advice can be applicable and helpful, anomalies and edge cases are arguably important yet challenging to catch using traditional machine learning approaches. The advantage of LLMs is that: (1) they have ample prior knowledge of statistics, insights on health, and common sense; (2) they can take into account the user's profile and other contextual information, such as gender, age, and habits. To generate meta-level insights, we used OpenAI's *GPT-4-turbo* model (temperature=0, max token=4096), which is an LLM model capable of receiving multi-modal input. We input the user's biography (provided by the user's demographic survey), the summary and correlation matrix of the data, and the plot of the hourly trends of heart rate and step count. We tried inputting the correlation matrix as a plot, but it resulted in consistent factual errors, so a numerical representation of the matrix was used instead. The system metaprompt instructs the LLM to generate at least 10 insights. For each insight, it needs to provide reasoning, assumptions, and explanations that make use of the data. The data sources need to be specific with values, and it must use a combination of different sources of data. After each insight, it needs to give a score between 0-10 on how likely the insight is to be the most important factor affecting sleep quality. An example output of this step is shown in Figure 2.

## C. Conversational Interface Design

The conversational interface is a text-based chatbot on a web browser that can be accessed on a phone or a laptop. The interface offers an interactive way to understand the data via a summary of data, discussions of implications, and answers to questions. The conversation is driven by an LLM which is prompt-tuned to focus on unique and personal trends and insights (Figure 1). We again used OpenAI's *GPT-4-turbo* model (temperature=0, max token=4096) but with a different system metaprompt. The model takes in the pre-generated insights and summary of the week of data as inputs. The system metaprompt of the LLM has a few critical components: **role**: defines the character of the LLM and its high-level role in the conversation; **data**: describes the expected input, including the person's biography, the summary of Fitbit data for the time period of interest, the correlation matrix, and health data trends; **communication style**: specifies a concise language style, avoiding overly technical jargon. **task**: ensures the LLM encourages users to explore all insights by suggesting relevant questions; **opening format**: grounds the conversation with a self-introduction, an overview of the data, derived insights, and three follow-up questions to guide user exploration. **caution**: anticipates and mitigates malicious or unintended uses of the LLM, such as off-topic questions; An example of the conversation is in Figure 1.

## V. USER STUDY

The focus of the work is not on the efficacy of LLMs' prediction capability but on motivating people to adopt more specific health behaviors as a result of the personalized interaction and interpretation of their wearable data. Our study is not designed to measure health outcomes; Rather, we investigate if our system can motivate users to adopt more specific health behaviors, which are linked to achieving positive outcomes. To that end, we implemented and evaluated our system in real-world settings where actual users interacted with our system using their wearable devices and personal data.

### A. Procedure

Participants were asked to wear the Fitbit for minimally a week, including during sleep. They completed a demographics survey and a pre-survey that asked about their understanding of their data and goals after using the Fitbit App. The survey breakdown is detailed in the later section. Once participants had at least a week of data, they exported and shared their Fitbit data with the experimenters. Their raw data was securely stored and never shared with any third-party systems, including the LLMs. They then interacted with a version of our system depending on which condition group they were randomly assigned to. They needed to complete at least 10 exchanges with the chatbot. Their chat conversations were logged and shared with the experimenters. Finally, participants completed a post-survey about the interface with the same questions as the pre-survey. Participants received $15 for completing the study, which was approved by the institution's IRB.

### B. Conditions

The study has 3 between-subject conditions: **Placebo (C1)**: Chat with an off-the-shelf LLM with no personal information; **Control (C2)**: Chat with an LLM that has access to a summary of their Fitbit data; **Intervention (C3)**: Chat with an LLM that has access to their Fitbit data summary, insights on how their data correlate, and generated follow-up questions that guide

the user through the insights. Note that the placebo group was still asked to share their Fitbit data, despite their summarized data never being provided to the conversational interface.

### C. Participants

50 Participants were recruited through university mailing lists, 5 participated in the pilot study, and 21 did not complete the full study and were excluded from the data analysis. This left us with 24 participants, 8 for C1, 8 for C2, and 8 for C3. The sample population has a mean age of 29.09 (SD=8.50). 12 identified as male, 12 as female. All have used a smartwatch before but may not be a Fitbit compatible watch. For consistency, we gave those who own a different type of smartwatch a Fitbit watch to wear for a week. Participants must not have any serious health or sleep concerns as our system should not provide medical diagnosis or advice. 77% typically use the Fitbit app at least once a day, 43% use LLM-based systems more than once a day, and 74% got full scores on the Cognitive Reflection Test [18], where a higher score indicates individuals' ability to suppress an intuitive and spontaneous wrong answer in favor of a reflective and deliberative right answer. In our study, this test assessed participants' acceptance of statistical explanations as opposed to adhering to prior beliefs.

### D. Hypotheses and Measurements

Below are the hypotheses and the corresponding metrics used to measure the effectiveness of the different conditions in four outcomes of interest. The pre-survey focuses on outcomes as a result of using the Fitbit App. The post-survey contains an identical set of questions as the pre-survey to measure the difference in the outcome after interacting with the chatbot.

- **H1**: C3>C2>C1 in improving individual's **understanding** of their data. Measured by: 7 qualitative questions, each followed by a quantitative self-rated confidence score, and 1 quantitative rating of the interface.
- **H2**: C3>C2>C1 in making individuals feel **motivated** to improve their sleep. Measured by: 1 qualitative question, and 3 quantitative ratings of the interface.
- **H3**: C3>C2>C1 in helping individuals form **actionable** goals to improve their sleep. Measured by: 1 qualitative question, and 3 quantitative ratings of the interface.
- **H4**: C3>C2>C1 as a more **personalized** interface. Measured by: 2 quantitative ratings of the interface.

### E. Analysis

For the quantitative results, we treat the mean of the aggregated 7-point Likert scores within each category as a continuous variable. We used a linear mixed effects (LME) model (lme4 package in R [19]) to account for the nested data structure, namely each subject has 2 observations: pre-survey and post-survey. We used the random intercept model, allowing each subject to have a unique intercept. The predictors of interest are **Test** (pre vs post) and **Condition** (placebo, control, and intervention), and we control for *AI literacy*, *Fitbit use frequency*, and *cognitive reflection test*. Intuitively, the four outcomes should not be independent as outcomes from

the same person have the same underlying determinants (e.g., one's motivation to improve their sleep could be dependent on how much they understand their data). However, we do not model the correlations among the outcomes at this point. Thus, we fit a random intercept linear mixed effect model for each outcome separately. Since our hypotheses need to compare three pairs of the conditions, and the LME model only compares two pairs (Control and Placebo, Intervention and Placebo), we conducted an post-hoc pairwise comparison using the emmeans package in R [20], which produces an adjusted p-value.

We also open-coded and thematically clustered qualitative questionnaire responses and conversation logs to extract trends. Specifically, we compared responses to the knowledge questions and action plans before and after interacting with the chatbot. We also compared the post-survey action plans against the conversation log. The hypothesis is that the conversation content has a positive influence on one's knowledge about their data, ability to generate actionable plans, and confidence in their knowledge and action plans.

## VI. Quantitative Results

Overall, interacting with a chatbot in addition to using the Fitbit app increased users' understanding. Specifically, the post-hoc pairwise comparison reveals that both Control and Intervention groups had a statistically significant increase in *understanding* (estimate=1.28 and 1.05 respectively, p<.01 and p=.02 respectively) (Figure 3). Comparing the amount of change post-interaction between the conditions, the LME model and pairwise comparison show that the Control group had a significantly greater increase than the Placebo group (estimate=1.01, p=.03) in *understanding* between the post- and pre-survey results (Figure 3), while the difference between Intervention and Placebo groups approached significance (estimate=0.77, p=.08) (Table I).

On the other hand, interacting with a generic chatbot that has no personal data or tailored insights felt *less personalized* than using the Fitbit App, whereas the full PhysioLLM system was rated the highest for this category (Figure 3). The pre-survey rating varied between conditions, so the pre-post interaction differences were not significant between conditions. The LME model did not reveal any statistical significance for the fixed or interaction effects. Similarly, the Placebo group rated the generic chatbot lower for outcomes *actionable* and *motivation* compared to using the Fitbit App alone, and the full PhysioLLM system was again rated the highest for both outcomes (Figure 3). Comparing between conditions, the LME model shows the Control group had a significantly greater increase in supporting *actionable* goals than the Placebo group (estimate=1.75, p=.03) (Figure 3, Table I).

## VII. Discussion

Combining numerical results and trends extracted from the qualitative results, we now discuss the system's performance in achieving our design goals.

**Comparing understanding pre- and post-interaction**. As mentioned earlier, the quantitative data shows the control and

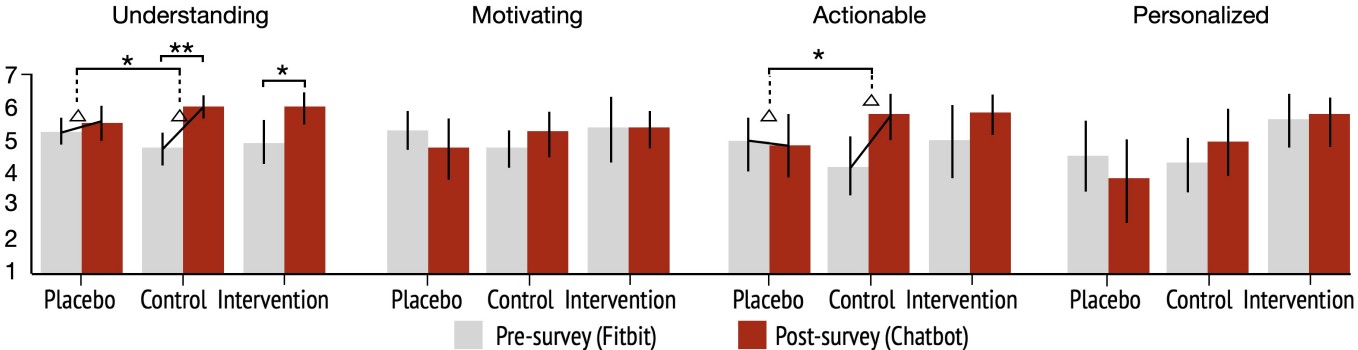

Fig. 3. Barplots of Likert-scale ratings. Higher ratings are better. Error bar: SE. *:p<.05,**:p<.01 $\Delta$: difference between pre- and post-survey.

intervention groups had a significant increase in confidence in their understanding of the data (Figure 3). Qualitative results revealed further that there was an increase in detail and clarity in post-survey responses. When asked if they knew what certain terminologies mean and their influence on sleep, there was a decrease in the number of "no" responses in the post-survey. For instance, many participants initially "vaguely" understood various sleep stages, but later described sleep's "importance for memory, emotions, other health regulation" (P202) in the post-survey. There were also several misconceptions before the interaction with the chatbot, and the responses of participants in control and interaction groups indicate a more comprehensive understanding afterward. For example, P34 initially only knew that "REM is when dream happens," but stated after the interaction that "REM (sleep) helps with memory & creativity and deep (sleep) is for restorative sleep"

while citing specific percentages of sleep stages. This was also seen with HRV knowledge, where P46 described that for them, "higher HRV is correlated with better sleep," whereas they initially had not heard of HRV.

When asked about what they thought had the most impact on their sleep, participants in the placebo group answered similarly in the pre- and post-survey, whereas those in control and intervention groups were able to pinpoint that physical activity during the day significantly affects their sleep (P32, P46, P402). Furthermore, participants were more specific about the timing of the activities. For example, in pre- and post-survey, participants mentioned that caffeine can make it harder to fall asleep, but post-study responses more frequently mention that "caffeine intake close to bedtime decreases sleep quality" (P33). Similarly, most participants conclusively stated in the post-survey that exercise leads to better sleep, an improvement from the varied and uncertain responses in the pre-survey.

**Comparing goals with conversation content**. We were also interested in whether the interaction with the chatbot led to more personal and actionable goals. Quantitative results show that the control and intervention groups rated the interface as more personalized and relevant and that they are more confident in their ability to use their health data to improve their sleep. The quantitative survey asked participants to list three goals and explain why and how they want to achieve these goals. In both the pilot study and full study, participants adapted their goals based on chatbot feedback. In particular, participants in the intervention group related daily behaviors with specific sleep outcomes based on insights provided by the system. For example, some goals were to "reduce stressful activities late at night" so they can "go to sleep at a more consistent time" (P40) or to have "more regular medium intensity exercise" for "better sleep and HRV" (P45). In contrast, participants in the placebo group had more personal goals, with vaguer explanations and reasoning on why they wanted to achieve them.

**Personalization**. Overall, participants had positive interactions and thought that conversations with the chatbot were "personalized" and "engaging." However, a few thought the chatbot was not personalized even though suggestions explicitly mention individual data such as steps per day or hours of sleep. A possible explanation may be that the health insights provided are well-known, making the chatbot responses appear

| Fixed Effects | Estimate | SE | df | t | p |
|---|---|---|---|---|---|
| **Understanding** | | | | | |
| (Intercept) | 3.84 | 0.73 | 21 | 5.24 | <0.001*** |
| Test | 0.27 | 0.30 | 21 | 0.91 | 0.37 |
| Control | -0.37 | 0.36 | 18 | -1.03 | 0.32 |
| Intervention | -0.34 | 0.35 | 18 | -0.94 | 0.35 |
| Test:Control | 1.01 | 0.43 | 21 | 2.37 | 0.03* |
| Test:Intervention | 0.77 | 0.43 | 21 | 1.82 | 0.08 |
| **Motivating** | | | | | |
| (Intercept) | 4.75 | 1.14 | 21 | 4.18 | <0.001*** |
| Test | -0.54 | 0.46 | 21 | -1.16 | 0.26 |
| Control | -0.32 | 0.56 | 18 | -0.57 | 0.58 |
| Intervention | -0.15 | 0.56 | 18 | -0.27 | 0.79 |
| Test:Control | 0.96 | 0.66 | 21 | 1.45 | 0.16 |
| Test:Intervention | 0.58 | 0.66 | 21 | 0.88 | 0.39 |
| **Actionable** | | | | | |
| (Intercept) | 4.21 | 1.29 | 21 | 3.27 | <0.001*** |
| Test | -0.16 | 0.54 | 21 | -0.31 | 0.76 |
| Control | -0.60 | 0.63 | 18 | -0.94 | 0.36 |
| Intervention | -0.25 | 0.64 | 18 | -0.39 | 0.70 |
| Test:Control | 1.75 | 0.76 | 21 | 2.31 | 0.03* |
| Test:Intervention | 0.96 | 0.76 | 21 | 1.26 | 0.22 |
| **Personalized** | | | | | |
| (Intercept) | 3.86 | 1.62 | 21 | 2.39 | 0.03* |
| Test | -0.69 | 0.54 | 21 | -1.25 | 0.22 |
| Control | -0.13 | 0.75 | 18 | -0.17 | 0.86 |
| Intervention | 1.10 | 0.76 | 18 | 1.45 | 0.17 |
| Test:Control | 1.38 | 0.78 | 21 | 1.77 | 0.09 |
| Test:Intervention | 0.81 | 0.78 | 21 | 1.05 | 0.31 |

TABLE I
RESULTS OF THE LINEAR MIXED EFFECTS MODELS TO TEST OUR HYPOTHESES. *Test* REPRESENTS FIXED EFFECT PRE- VS POST-SURVEY. ":" INDICATES AN INTERACTION BETWEEN TWO EFFECTS, I.E., THE BETWEEN-CONDITION COMPARISONS IN THE HYPOTHESES. PLACEBO IS THE BASELINE. *:P<.05,**:P<.01,***:P<.001.

more generic. Nonetheless, participants felt that the interface focused them on the relevant information.

## VIII. Preliminary evaluation with sleep experts

In addition to the study with laypeople, we worked with sleep experts to investigate whether our system can generate insights that are otherwise "hidden" in the data, such as potential causality and relationships between the user's activities and various aspects of sleep. Two sleep experts, B and J, were independently interviewed using an experimenter's personal data as a case study. They were presented with the same input (biography, summary, correlation, and trends) given to the LLM and asked to generate insights without additional context. They then reviewed the LLM's generated insights and the system's responses via its interface. Below, we summarize the main insights from both interviews.

**Comparison between LLM and human expert insights.** We compared how human experts and PhysioLLM approached the provided information. Both experts focused on big-picture data to assess the user's sleep health, whereas the LLM concentrated on data correlations. Some insights generated from the correlation matrix were similar between the LLM and the experts. However, experts found some correlations unexpected and counterintuitive, such as an increase in sedentary minutes correlating with a higher percentage of deep sleep. The LLM justified this by suggesting it "could be due to the body's increased need to recover from activity." In contrast, the experts dismissed this correlation, noting that the step count and activity minutes indicated the person did not engage in activities intense enough to require such recovery.

**Expert opinion on insights.** Overall, the system provided reasonable and correct explanations. Most of the explanations that experts found surprising stemmed from unexpected correlations in the data. The LLM tends to over-index the correlation values. Experts noted that correlation significance should be adjusted for the small data sample and redundant data categories. They suggested reducing comparisons by combining related values, such as aggregating different activity levels into a single value.

**Expert opinion on generated feedback.** Expert J took the perspective of the user and thought it gave "good suggestions on the practical side," while expert B took the perspective of a medical professional. Expert B remarked that since some insights might be based on spurious statistics, the model should provide more modest comments rather than sounding certain. While acknowledging occasional over-interpretation by the model, Expert J believed that "the explanations may not matter," as users primarily seek actionable advice, such as avoiding overexertion and not exercising close to bedtime.

## IX. Limitation and Future Work

**Limitations of sensor data.** We assume most people follow a weekly routine, so we choose a week of data as the range of input data. Some correlation values can be counter-intuitive due to the short time window of data. In addition, several different health conditions can cause the same changes in sensor readings. For example, heart rate variability can be low due to stress, or because one has an infection. Because the data are inherently ambiguous, the system should not try to provide specific diagnoses based on the data, rather it should suggest testable hypotheses to the user which they can try to identify the root causes.

**Limitations of insights.** The current implementation relies on GPT's prior knowledge during training. This is acceptable as prior work has shown that the zero-shot GPT-4 can have 84% accuracy when answering medical licensing exams [10]. A fine-tuned GPT for medical diagnosis can improve the accuracy and comprehensiveness of the system. The way the insights are presented could also be more diverse. Some participants wished they were given more visuals, such as graphs to represent the data the chatbot is referencing. In the future, the conversational interface can be directly integrated into the companying app, and the chatbot can reference the graphical representations in addition to the textual insights.

**Safety, privacy, and ethics.** The system has embedded counter-action prompts to prevent abusive uses of the system that are beyond the system's capabilities and intended uses, but further tests on the robustness of the safety prompt are needed. The outcome of the generation should be factually accurate, especially in the domains of personal health. Mistakes such as Google's AI search feature suggesting people eat rocks[4] highlight the challenge of making the LLM factually grounded. However, not all mistakes are obvious. The human-sounding outputs of the LLM systems are worrisomely persuasive. We made sure users knew the system was not allowed to provide medical diagnoses and advice, and that the system should acknowledge its limitations. Last but not least, health and activity data is sensitive information. By design, we ensured that no raw data was sent to the LLM, and we de-identified all data and survey results. Using a local model would be even more secure, and there are increasingly more lightweight models that run on local hardware. However, these models perform worse on benchmarks like MMLU and would need further fine-tuning. Thus, we chose GPT-4-turbo for its state-of-the-art performance in reasoning and accuracy[5].

**Participant pool.** Our participant pool is small and leaning towards people with higher education. We controlled for factors like AI knowledge, prior app usage, and cognitive thinking style and found no significant effects, so there was no noticeable difference due to demographics. The participant group was recruited with some interest in improving their sleep but most had no specific sleep issues. This reduced the likelihood of our system discovering findings that were significantly different from common knowledge and suggested actions that could result in drastic behavior change. In the future, we hope to work with a broader user group with varying degrees of exposure to AI and with more diverse sleep patterns.

**Just-in-time assistance.** Our system allows the user to reflect on recent but historical data. A proactive, always-on

---

[4]https://www.bbc.com/news/articles/cd11gzejgz4o
[5]https://openai.com/index/hello-gpt-4o/

system could suggest and anticipate physiological states to help individuals take preventive measures.

**Generalizability of our approach of time-series data**. Our earlier experimentations showed the importance of an accurate and interpretable representation of the input data to the LLM. One unique challenge of interfacing with time-series data is the alignment of data sampled at different frequencies, different segments of the day (e.g., only during sleep or active exercises), and potentially different time zones). In addition, some data should be aggregated (e.g., step count at the end of the day) versus averaged (e.g., heart rate at each hour). These nuances depend on the contextual meaning of the data. This challenge motivated our approach of adding a data processing step before interfacing with the LLM.

## X. CONCLUSION

We introduced PhysioLLM, a novel system that addresses the question of how to provide personalized health insights from individuals' wearables. The system orchestrates multiple LLMs and non-LLM modules to generate reliable, personal, and insightful outputs. Our user study with 24 Fitbit watch users demonstrates that PhysioLLM outperforms both the Fitbit App and a generic LLM chatbot in facilitating a deeper, personalized understanding of health data and supporting actionable steps toward personal health goals. Despite limitations, such as handling the randomness and unknowns in the data and contexts, the adaptability of our system ensures beneficial and personalized suggestions. Our system uses an off-the-shelf, general-purpose LLM so it has limited expert health knowledge; integrations of fine-tuned specialized LLMs with our system will further improve the quality of the insights. As LLM-based conversational systems become widely integrated with health apps, our insights are eminently important for providing appropriate responses and enabling users to query and discover insights. Anecdotally, some participants reported deeper reflections about their sleep and adjusted daytime activities informed by the interactions with our system, which shows the promise of this system in nudging people towards positive behavior change and merits future study. The significance of this work lies in its potential to turn general-purpose LLMs into personal intelligence by contextualizing AI-enabled conversational chatbots with time-series, personal data. We envision that this system allows individuals to better understand how their body functions and the consequences of actions, thereby making the internal and invisible visible.

## ACKNOWLEDGMENT

Statistical support was provided by data science specialist Jinjie Liu at the Institute for Quantitative Social Science, Harvard University. The authors would like to thank Dr. Jing Zhang and Dr. Robert Stickgold for their expert opinion, Leyla Omeragic Buljina for supporting subject recruitment, Noah Jones, Boyu Zhang, Prof. Roz Picard for their advice, and finally all study participants for their time.

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
