# OpenReview forum: "PhysioLLM: Supporting Personalized Health Insights with Wearables and Large Language Models"
_IEEE.org/EMBS/BHI/2024/Conference — IEEE BHI'24_

### Official Review · Reviewer_TEX4 · 2024-08-06
**The paper presents PhysioLLM, a system leveraging large language models to provide personalized health insights by integrating physiological data from wearables. The system aims to enhance user understanding of health data and support actionable health goals, with a focus on improving sleep quality. The study demonstrates that PhysioLLM outperforms standard wearable apps in delivering personalized health insights.**

**Overall Rating:** 8
**Confidence:** 3

**Other Quality Metrics:**

(a) Clarity of writing: Great
(b) Clinical Significance: Good
(c) Methodological Novelty: Excellent
(d) Experiments and Results: Good

**Questions For The Authors:**

Can the system be easily adapted to focus on other health metrics besides sleep quality? If so, what are the potential challenges? It would interesting to explore the potential to integrate such methodology into the field of human movement analysis in a way that clinical questionnaires might be mixed with mobility outcomes data from lab (or daily life) assessment. Happy to discuss this further.

How does PhysioLLM ensure user privacy and data security, especially with sensitive health data? It might be relevant to expand this part.

Were there any significant differences in the effectiveness of PhysioLLM between different demographic groups within your study sample?

Did the authors think of using any data standardization approach to have more harmonized/structured data?

**Strengths:**

Innovative Integration: Combines wearable data with LLMs for personalized health insights.
Comprehensive Analysis: Offers detailed statistical analysis and trend discovery beyond what commercial apps provide.
User-Centric Design: Focuses on improving user understanding and actionable health goals.
Validation: Includes user studies and expert reviews to substantiate the system's effectiveness and reliability.

**Summary Of The Paper:**

The paper introduces PhysioLLM, an interactive system designed to integrate wearable-collected physiological data with large language models to generate personalized health insights. Unlike commercial health apps, PhysioLLM conducts comprehensive statistical analyses to uncover correlations and trends in user data. A case study focusing on sleep quality illustrates the system's effectiveness. A user study with 24 participants shows that PhysioLLM improves understanding and motivation compared to a standard Fitbit app and a generic LLM chatbot. Insights from sleep experts validate the system's generated advice but suggest improvements to handle sparse data and reduce overemphasis on correlations.

**Weaknesses:**

Overemphasis on Correlations: The system may overstate the importance of certain correlations, which can mislead users.
Sparse Data Handling: Needs improvement in handling sparse data to ensure accurate insights.
Generalization: While sleep quality is a significant focus, the system's effectiveness across other health metrics is not deeply explored.
User Study Size: The sample size of 24 participants, while sufficient for initial insights, may not be enough representative.

---

### Official Review · Reviewer_M1gA · 2024-08-18
**A preliminary investigation that could improve on core contribution, methodology, and results/evaluations**

**Overall Rating:** 5
**Confidence:** 4

**Other Quality Metrics:**

- Clarity of writing: **fair**
- Clinical Significance: **good**
- Methodological Novelty: **fair**
- Experiments and Results: **fair**

**Questions For The Authors:**

- **LLM interaction**: How were the demographics and physiological data fed into the LLM? What _bespoke function_ was used to help LLM to generate the desired format of health insights?

**Strengths:**

- The manuscript presented an innovative idea of proposing a pipeline that could interpret and explain physiological data to users with LLMs.
- The manuscript provided good visuals to complement its explanations.

**Summary Of The Paper:**

Utilization of large language models (LLMs) in the health domain could offer improved personalized advising and insights about health habits to users. On the other hand, wearable health monitors such as smart watches could collect physiological data representing the user's daily well-being. However, explorations on how LLMs could interpret wearable physiological data and provide personalized, precise explanations and advice on well-being remained sparse. This manuscript utilized improving sleep quality as a case study and proposed PhysioLLM to understand wearable data, explain it to users, and provide advice on improving sleep quality. It evaluated the proposed pipeline on a pilot study with 24 self-recruited participants and interviews with 2 sleep experts.

**Weaknesses:**

- **Core Limitation and Solution**: The manuscript's description of the core medical challenge and how its work could overcome it was unclear. Is there a lack of an automatic physiological data interpretation tool? Is it about the lack of automatic feedback tools for improving personal health? Or both? Why would previous works, such as PH-LLM, Health-LLM, or non-AI tools fail and PhysioLLM would triumph?

- **Methodological Innovation**: The manuscript's technical innovation was unclear. The manuscript touched on each aspect of the framework but failed to provide enough details to explain its implementation. How did LLM interact with wearable data? How did LLM interact with users? Was there a mechanism to evaluate or prevent the model to hallucinate or speak general information?

- **Evaluation on Specificity**:The manuscript lacked evaluations of its efficacy of predicting health concerns, particularly on its specificity and sensitivity. Even if the pilot study collected no real sleep conditions (labels), a side-by-side evaluation could be done by comparing the LLM's response to the same prompt, one with physiological and demographic data, and one without.

- **Significance of Findings**: The manuscript's evaluation was performed on a dataset of only 8 participants in each control group, which greatly limited the statistical significance of its evaluation. Moreover, the recruitment was done through university mailing and introduced implicit bias on age, education, and the ability to use digital technologies (tech-savviness). All of these factors could further diminish the impact of the study.

---

### Official Review · Reviewer_N2qt · 2024-08-19
**PhysioLLM develops Personalized Insights for User Health**

**Overall Rating:** 7
**Confidence:** 4

**Other Quality Metrics:**

a) Clarity of Writing : Excellent
b) Clinical Significance: Fair
c) Methodological Novelty: Great
d) Experiments and Results: great

**Questions For The Authors:**

1. What does an insight from this system look like? Is it simply identifying features with the highest/lowest values in the correlational matrix?
2. Does the system produce the same insight for users every time a single user uses it?
3. How useful do you think these insights are? To what extent can they be used to make meaningful, actionable changes in the users' lives?

**Strengths:**

"The concept of using AI-based systems to analyze physiological data and derive valuable insights about the user is very promising. This paper discusses recent trends in computation, such as:

1. Health tracking using wearables
2. LLM's capability to comprehend context and generate useful insights

I believe the study's design is robust, and the tested hypotheses are considerate in developing an LLM that can effectively provide valuable and insightful responses to the user

**Summary Of The Paper:**

This paper introduces PhysioLLM, a system based on LLMs designed to provide personalized reports on a user's health.

With the increasing use of fitness trackers, users are now collecting data from various sensors to monitor physiological functioning. However, for the average user, extracting meaningful insights from this data to improve their health outcomes can be challenging.

PhysioLLM aims to address this challenge by generating actionable insights about the user's health outcomes. It does this by processing the data and utilizing an LLM to produce these insights.

For the study, the LLM was provided with a user biography, a summary of trends and correlation matrix, as well as plots of hourly trends, and was tasked with generating insights about sleep. The performance of the LLM in producing effective insights was evaluated.

**Weaknesses:**

1. I am concerned about the anonymization of user data. The paper claims that user privacy has been preserved, but it uses the GPT-4 turbo model to generate insights about user data. GPT 4t can be accessed using an API, which means that data is sent to a third-party server. Perhaps using a open source local GPT might help preserve user data more effectively.

2. More explanation is needed for Fig 3, which details how the study was conducted.

3. What constitutes a good insight? The study participants had positive interactions with the LLM, but it's unclear how these insights translated into useful health outcomes for the users.

---

### Decision · Program_Chairs · 2024-09-23

Accept